# Bidirectional Associations between Daytime Napping Duration and Metabolic Syndrome: A Nationally Representative Cohort Study

**DOI:** 10.3390/nu14245292

**Published:** 2022-12-12

**Authors:** Jinqi Wang, Zhiyuan Wu, Xiaohan Jin, Rui Jin, Ze Han, Haiping Zhang, Zongkai Xu, Yue Liu, Xiuhua Guo, Lixin Tao

**Affiliations:** 1Beijing Municipal Key Laboratory of Clinical Epidemiology, Department of Epidemiology and Health Statistics, School of Public Health, Capital Medical University, Beijing 100069, China; 2Department of Public Health, School of Medical and Health Sciences, Edith Cowan University, Perth, WA 6027, Australia

**Keywords:** daytime napping duration, metabolic syndrome, metabolic syndrome remission, bidirectional association, cohort study

## Abstract

Background: We aimed to examine the bidirectional associations between daytime napping duration and metabolic syndrome (MetS). Methods: Using data from the China Health and Retirement Longitudinal Study from 2011 to 2015, modified Poisson regression models were performed to explore the longitudinal associations of baseline napping duration with the occurrence and remission of MetS. Generalized estimating equation was conducted to explore the association between baseline MetS status with subsequent changes in daytime napping duration. Cross-lagged panel analysis was performed to further verify their bidirectional relationships. Results: During the four-year follow-up, among 5041 participants without MetS at baseline, extended naps were significantly associated with MetS occurrence, compared with non-napping. This association was only significant in individuals with adequate night-time sleep duration or good sleep quality of the 2898 participants with MetS at baseline. Excessive napping duration may be not favorable for MetS remission especially for adequate night-time sleepers. With respect to reverse associations, baseline MetS status significantly increased the napping duration during the subsequent follow-up period. Finally, there were significant bidirectional cross-lagged associations between napping duration and MetS severity score after adjusting for all covariates. Conclusions: Our study indicates bidirectional relationships exist between daytime napping duration and MetS. Interestingly, longer napping duration was detrimental to cardiometabolic health only in those with sufficient night-time sleep duration or good sleep quality.

## 1. Introduction

Metabolic syndrome (MetS) is characterized by a set of cardiometabolic risk factors, including abdominal obesity, hypertension, hypertriglyceridemia, hyperglycemia, and lowered high-density lipoprotein-cholesterol (HDL-C) [1]. MetS serves as a well-recognized, strong, independent predictor of type 2 diabetes mellitus, cardiovascular diseases, and all-cause mortality [1,2,3]. Therefore, it is particularly crucial to identify potential risk factors to prevent the incidence of MetS and promote its remission. Sleep-related factors, such as daytime napping, was considered to be modifiable behavioral factors affecting cardiometabolic health [4,5].

Daytime napping is a common behavior in many countries. A previous study [6] demonstrated that the prevalence of daytime napping was 57.7% among the Chinese elderly population. As an appropriate way to counteract the negative consequences of sleep debt [7,8], daytime napping was considered as a healthy lifestyle and could offer benefits such as memory consolidation, cognitive performance enhancement, and a boost in emotional stability [9,10]. Several randomized controlled trials suggested that daytime napping reversed the adverse effects of night-time sleep loss on the secretion of multiple hormones and cytokines [11,12]. However, daytime napping duration can range from minutes to over 1 h [8]. Accumulative evidence showed that excessive daytime napping was associated with many adverse outcomes, such as cardiovascular disease and all-cause mortality [8,13]. Some epidemiological studies have reported that longer daytime napping duration was related to MetS [4,7,14,15,16,17,18,19,20]. Considering the double-sided role of daytime napping, we inferred that the effects of napping duration on MetS may be modified by night-time sleep. The different associations mentioned above among those with different night-time sleep duration and quality still need further exploration. Additionally, most of the previous studies were limited due to the cross-sectional design or lack of representativeness of study populations. In addition, whether longer daytime napping duration impacts on the remission of MetS remains unclear. Thus, the associations between daytime napping duration and the occurrence or remission of MetS need more prospective evidence.

Notably, previous studies examining the relationship between daytime napping and MetS are typically undertaken with the assumption of a unidirectional relationship, such as “longer daytime napping duration may lead to future changes in MetS status”. However, there may exist a bidirectional influence between them. To our knowledge, no study has explored whether baseline MetS status and its components can predict the future changes in napping behavior. A bidirectional association between daytime napping and MetS warrants further clarifications, which will further elucidate their complicated underlying mechanism and provide new insight into their prevention.

The China Health Retirement Longitudinal Study (CHARLS) is a nationally representative aging cohort that involved daytime napping duration and MetS status at multiple waves [21]. This provided an opportunity to explore the potential bidirectional relationship between daytime napping and MetS. Considering the limitations of previous research, we used the CHARLS database and aimed to: (1) investigate the longitudinal association of baseline daytime napping duration with the occurrence and remission of MetS, and explore the above associations in different subgroups stratified by night-time sleep duration and quality; (2) examine the longitudinal associations of baseline MetS status and its components with subsequent changes in daytime napping duration; and (3) performe cross-lagged panel analysis to further verify their bidirectional relationship.

## 2. Materials and Methods

### 2.1. Study Design

CHARLS was conducted among Chinese adults aged 45 years or older [21], which is a prospective cohort collecting data on social and economic information, anthropometric and laboratory measurements, demographic characteristics, health-related behaviors, and health conditions. The baseline survey applied a four-stage, stratified, cluster probability sampling method from June 2011 to March 2012. A total of 17,707 participants from 150 county-level units distributed in 28 provinces of China were recruited at baseline and followed up every 2 years. Details about the data were previously described in an earlier publication [21]. The CHARLS data are available for the baseline survey in 2011 (wave 1), the first follow-up survey in 2013 (wave 2), and the second follow-up survey in 2015 (wave 3). Only in the wave 1 and wave 3 surveys did the participants undergo metabolic examinations.

### 2.2. Ethics

The protocols of CHARLS were approved by the Biomedical Ethics Review Committee of Peking University (IRB00001052-11015). All participants provided their signed informed consents.

### 2.3. Study Population

We identified three study sub-cohorts from CHARLS. The process for selecting participants in these three sub-cohorts and the purpose of sub-cohorts’ construction were as follows.

Of the 17,707 CHARLS participants at study baseline, we excluded subjects with missing information on age, gender, sleep-related variables, and MetS status at baseline. Participants diagnosed with cancer, malignant tumors, stroke, or heart disease were also excluded. A total of 13,821 participants who met the initial inclusion criteria were included for further sub-cohorts’ construction. The first sub-cohort was constructed to explore the longitudinal association between baseline daytime napping duration and the occurrence of MetS. Then, 8780 participants were further excluded if they were diagnosed with MetS at baseline, did not attend the follow-up survey, or lacked the required information to define MetS in 2015. Finally, 5041 participants without MetS were considered in the final analyses. The second sub-cohort was to examine the influence of baseline daytime napping duration on remission of MetS. A total of 10,923 participants were excluded if they were without MetS at baseline or without the required information to define MetS in 2015. Finally, 2898 participants with MetS were enrolled. The data used in the sub-cohort 1 and sub-cohort 2 were from the wave 1 and wave 3 surveys of CHARLS. The third sub-cohort was to test the association of baseline MetS with subsequent changes in daytime napping duration. Then, 2431 participants without at least one reassessment of daytime napping were further excluded. A total of 11,390 participants were included in the analyses of reversed associations, and the data used were from the wave 1, wave 2, and wave 3 surveys.

The study flow chart was illustrated in Appendix A and the time line of the study was presented in Appendix A.

### 2.4. Measurements of MetS Status

Data collection was performed by medically trained staff from the Chinese Center for Disease Control and Prevention. Self-reported past medical history and medication history (hypertension, dyslipidemia, diabetes) were obtained using structured questionnaires. Then, each person’s blood pressure was measured three times with an automated electronic device (OMRON Model HEM-7200; Omron Company, Dalian, China), and the average of the three measurements was taken as the final blood pressure. To measure waist circumference, the researchers placed measuring tape over the clothing around the waist at the level of the navel and recorded the measurement. Venous blood was collected from each participant with their fasting status recorded. Total Serum cholesterol (TC), triglycerides (TG), low-density lipoprotein cholesterol (LDL-C), and HDL-C were examined using an enzymatic colorimetric test. Blood glucose level was checked using the glucose oxidase method. Other details in the questionnaire, physical measurement, and blood collection were described elsewhere [21,22].

We defined MetS based on a joint statement in 2009 [23]. Participants who met ≥3 of the following 5 criteria were considered to have MetS: (1) waist circumference ≥ 90 cm for men and ≥85 cm for women; (2) systolic blood pressure ≥ 130 mmHg or diastolic blood pressure ≥ 85 mmHg or self-reported hypertension or using antihypertensive drugs; (3) fasting plasma glucose(FPG) ≥ 5.6 mmol/L or self-reported diabetes or the use of diabetes medication; (4) reduced plasma HDL-C (<1.0 mmol/L for men and <1.3 mmol/L for women) or specific drug treatment for this lipid abnormality; and (5) elevated plasma TG (≥1.7 mmol/L) or specific drug treatment for this lipid abnormality.

The remission of MetS was defined as no longer fulfilling the diagnostic criteria for MetS after four-year follow-up in 2015. MetS severity score was defined as the number of MetS components present, and the range is from 0 to 5 [24].

### 2.5. Assessment of Sleep-Related Variables

Daytime napping duration was appraised using self-reported questionnaires which asked: “During the past month, how long did you take a nap per day on average?”. Night-time sleep duration was assessed by the following question: “During the past month, how many hours of actual sleep did you get at night (average hours for one night)? This may be shorter than the number of hours you spend in bed.”. Night-time sleep quality was assessed by asking the question “my sleep was restless”, and participants selected the following four options based on their feelings and behaviors in the last week—“rarely or none the time (<1 day)”; “some or a little of the time (1–2 days)”; “occasionally or a moderate amount of the time (3–4 days)”; and “most or all of the time (5–7 days)” [25,26]. These questions had high validity and reliability for measuring sleep-related variables [27].

Participants were categorized into four napping groups: non-nappers (0 min/day); short nappers (≤30 min/day); moderate nappers (30–90 min/day); and extended nappers (>90 min/day) [6,28]. The optimal duration of sleep for adults is 7 or more hours per night according to a recent consensus recommendation developed by the American Academy of Sleep Medicine (AASM) and the Sleep Research Society (SRS) [29]. Therefore, we defined participants who slept more than or equal to 7 h at night as adequate night-time sleepers. On the contrary, participants who slept less than 7 h per night were defined as insufficient night-time sleepers. Participants were also divided into two subgroups based on their sleep quality. The option “rarely or none of the time (<1 day)” for “my sleep was restless” was represented as a good sleep quality, and other options were represented as a fair/poor sleep quality [25].

### 2.6. Assessment of Covariates

Several covariates were included in our analyses.

#### 2.6.1. Sociodemographic Characteristics

Sociodemographic characteristics included age, gender (male and female), living residence (rural and urban), marital status (married and others), and educational level (no formal education and elementary school or above).

#### 2.6.2. Health-Related Factors

Health-related factors included self-reported smoking status (non-smoker, former smoker, and current smoker) and drinking status (drink more than once a month, drink but less than once a month, and never). Depressive symptoms were assessed using the 10-item Center for Epidemiological Studies Depression Scale (CES-D-10), which has 10 questions with a scale of four points, and ranges from 0 to 30. A higher score indicates more depressive symptoms, and a cutoff score ≥ 10 was used to identify the respondents who had significant depressive symptoms [30,31]. Information on antidepressant treatment and the usage of sleeping pills was also collected by self-reports. Body mass index (BMI) was calculated as weight in kilograms divided by height in meters squared. The information of high sensitivity C-reactive protein (hsCRP), LDL-C and serum uric acid (SUA) were obtained through metabolic examination. Finally, physical activity (PA) was assessed by the questionnaires and participants reported the number of days and the time spent each day for three activity types (mild, moderate, and vigorous) in a usual week. “Mild”, “moderate”, or “severe” PA is defined as the corresponding activity type that has three or more days per week and lasts at least 10 min each time [32]. According to the highest PA intensity they reported, participants were divided into “none”, “mild”, “moderate”, and “vigorous” PA [32].

### 2.7. Statistical Analysis

Baseline characteristics were presented as the mean (standard deviation, SD), median [interquartile range, IQR], or number (percentage), as appropriate.

Firstly, to estimate the associations of baseline daytime napping duration with the occurrence and remission of MetS, a modified Poisson regression for binary outcome data with a log link function and robust error variance was performed. Daytime napping duration was analyzed as a categorical variable and a continuous variable. We evaluated the longitudinal dose-response relationship between daytime napping duration, as continuous change, and the incidence or the remission of MetS using restricted cubic splines (RCS) with 4 knots (at the 5th, 35th, 65th, and 95th percentiles). To adjust for potential confounding factors, two models were established as follows: Model 1 was adjusted for age and gender; Model 2 was further adjusted for living residence, marital status, educational level, smoking status, drinking status, depressive symptoms, night-time sleep quality, night-time sleep duration, SUA, LDL-C, and hsCRP. The adjusted relative risks ratio (aRRs) and 95% confidence intervals (CIs) were reported. The above-mentioned associations were also assessed in subgroup analyses by night-time sleep duration (night-time sleep duration ≥ 7 h, night-time sleep duration < 7 h), and sleep quality (good sleep quality, fair/poor sleep quality), adjusting for the same potential confounders. Then, we also investigated the relationships between daytime napping and the occurrence or remission of MetS components.

A series of sensitivity analyses was conducted to test the robustness of the results in this part: (1) we further adjusted for PA and multiple medications therapy (including antihypertensive agents, hypoglycemic agents/insulin, lipid-lowering agents, and sleeping pills/anti-depressive treatment) in model 3 and model 4; (2) propensity scores for the napping categories were generated using multinomial logistic regression. The above relationships were repeatedly estimated from the inverse probability weighted logistic regression models; (3) we re-evaluated the associations after changing the definition of MetS. This definition of MetS is based on the criteria of International Diabetes Federation (IDF) [33] and shown in the Appendix A; (4) we used ordinal logistic regression to estimate associations between daytime napping duration and MetS severity score. The adjusted model included Model 2 variables plus MetS severity score at baseline; (5) we analyzed the above associations in subgroups, according to age, gender, and depression status.

Secondly, to assess the associations of baseline MetS status, MetS severity score, and MetS components with subsequent changes in daytime napping duration over a period of 4 years, we used a generalized estimating equation (GEE) model and further adjusted for the napping duration at baseline in Model 2. To evaluate the stability of the observed results, sensitivity analyses were conducted as follows: (1) we further included PA, antihypertensive agents, hypoglycemic agents/insulin, lipid-lowering agents and sleeping pills/anti-depressive treatment as adjustment factors; (2) napping change was calculated by differences in napping duration between 2011 and 2013, or 2011 and 2015, which were used as the secondary outcomes and multiple linear regression models were used to verify the above results; (3) subgroup analyses based on gender and age were performed.

Finally, in the third sub-cohort, 5494 individuals had complete data on napping and MetS in both wave 1 and wave 3 surveys. A cross-lagged panel analysis was performed to further verify the bidirectional relationship between naps and MetS. Prior to cross-lagged analysis, the baseline and follow-up values of napping duration and MetS severity score were adjusted for all covariates by regression residual analyses and then standardized with Z-transformation. A test statistic Z was calculated to compare the coefficients of two-way path.

Two-sided *p* < 0 .05 was considered as statistically significant. Modified Poisson regression models were conducted using SAS statistical software Version 9.4, (SAS Institute Inc., Cary, NC, USA). Other statistical analyses above were performed with R software version 4.1.0.

## 3. Results

### 3.1. Baseline Characteristics

Table 1 summarizes the demographic and clinical characteristics of three sub-cohorts. Of the 5041 participants free of MetS at baseline in the first sub-cohort, during the four-year of follow-up, 1126 (22.3%) participants developed MetS. Of the 2898 participants with MetS at baseline in the second sub-cohort, 828 (28.6%) participants recovered after four-year of follow-up. Comparison of baseline characteristics according to napping duration are presented in Appendix A. Participants in the third sub-cohorts were grouped by MetS status and the differences of the characteristics are shown in Appendix A.

### 3.2. Association of Baseline Daytime Napping Duration with the Occurrence of MetS

For all participants without MetS at baseline, after step-forward adjustment for confounding covariates, the trend of a higher risk of MetS incidence for a longer daytime napping duration was similar across the two multivariable adjusted models (Table 2). In the fully adjusted model (Model 2), per-ten minutes increase in napping duration was significantly associated with an increased risk of MetS, and the aRR value was 1.013 (95% CI, 1.002–1.024). When napping duration was analyzed as categorical variables, we found that only extended nappers had a significantly higher risk of incident MetS compared with non-nappers (aRR, 1.216; 95% CI, 1.047–1.413). The J-shaped dose–response relationship between napping duration and the incidence of MetS is presented in Figure 1A, and we can see the relationship was significant only when the napping duration was greater than 90 min/day. Appendix A presents the aRRs and 95% CI for the incident MetS components. Longer daytime napping had a significant impact on the central obesity (aRR _extended nappers vs. non-nappers_, 1.204; 95% CI, 1.038–1.397; aRR _as a continuous variable_, 1.014; 95% CI, 1.003–1.025).

A subgroup analysis was performed based on night-time sleep duration and quality. As shown in Figure 2A, the significant association between extended naps and the occurrence of MetS was found in those with sufficient night-time sleep (aRR, 1.376; 95% CI, 1.139–1.676) and good sleep quality (aRR, 1.316; 95% CI, 1.073–1.615), compared with no napping. Significant associations were also observed when napping duration was treated as a continuous variable in both subgroups (aRR _with adequate sleep duration_, 1.022; 95% CI, 1.006–1.038; aRR _with good sleep quality_, 1.018; 95% CI, 1.001–1.034). On the contrary, no significant association between longer daytime napping and higher risk of incident MetS was observed in those with shorter night-time sleep or relatively poor sleep quality. The dose–response relationship between the daytime napping duration and the occurrence of MetS in different subgroups (Figure 1B–E) also confirmed the above results.

### 3.3. Association of Daytime Napping Duration with the Remission of MetS

For all participants with MetS at baseline, no statistically significant associations were detected between daytime napping and MetS remission in the two models (Table 2). Nevertheless, in the fully adjusted model, this association was found in participants with adequate night-time sleep duration (Figure 2B). Among those with sufficient sleep at night, with every increase of 10-min of napping duration, the likelihood of MetS remission was reduced by 2% (95% CI, 0.3–3.7%). Compared with no napping, extended naps may be not favorable for the remission of MetS (aRR, 0.754; 95% CI, 0.585–0.972). Conversely, short naps may promote MetS remission, but these associations were not significant (aRR, 1.086; 95% CI, 0.832–1.418). Appendix A shows the dose–response relationship between the napping duration and the remission of MetS, which also could confirm the above results. No adverse effects of extended naps were found in those with short night-time sleep duration and poor night-time sleep quality. Finally, we also examined the associations of napping duration with the remission of MetS components (Appendix A). Longer daytime napping, especially >90 min/day, was less likely to revert hypertension (aRR _extended nappers vs non-nappers_, 0.612; 95% CI, 0.429–0.873; aRR _as a continuous variable_, 0.968; 95% CI, 0.945–0.992). With respect to the other components of MetS, no statistically significant associations were demonstrated.

### 3.4. Association of Baseline MetS Status with Changes in Daytime Napping Duration

Prior to the main analyses, GEE was performed to take into account the association of changes over the follow-up time by including a term for time and the main effect of MetS-related variables, and a term for the interaction between them in fully adjusted models. However, both the term for time and interaction terms were not statistically significant. Hence, we focused only on the main effect of MetS. The results of the crude and fully adjusted GEE models are shown in Table 3. MetS significantly increased the duration of daytime napping during the subsequent follow-up period (*β*: 2.745; 95% CI: 1.360–4.130). When we assessed the associations between the number of MetS components (MetS severity score) and napping duration, the regression coefficient increases gradually as the MetS severity score value increases. Compared with those without abnormal component of MetS, the subsequent daytime napping duration increased by 7.053 min (95% CI: 3.435–10.670) among participants with five components of MetS. Analyses of the individual MetS components revealed that hyperglycemia, hypertriglyceridemia, hypertension, and central obesity were all significantly associated with future longer daytime napping duration. Central obesity shows the largest effect (*β*: 2.710; 95% CI: 1.309–4.111).

### 3.5. Sensitivity Analysis

Results of sensitivity analyses for unidirectional relationship of “baseline nap follow-up MetS” are presented below. Our findings were robust after further adjusting for PA and multiple drug therapy (Appendix A), using inverse probability weighted models (Appendix A) or changing the definition of MetS (Appendix A). Additionally, compared with non-nappers, extended nappers had higher MetS severity scores in both sub-cohort 1 and sub-cohort 2 (Appendix A). Similarly, no significant associations were found in those with insufficient sleep or poor sleep quality. Results of subgroup analyses by depression status, age and gender are presented in Appendix A. The associations between longer naps and incident MetS were significant in all subgroups by age and gender. However, such association was only observed among those with no depression but not among patients with depression. With respect to the inverse associations, the results remained almost consistent after further adjusting for PA and multiple drug therapy (Appendix A), or using the secondary outcomes (Appendix A). The associations were also statistically significant in all subgroups based on age and gender (Appendix A).

### 3.6. Cross-Lagged Panel Analysis

Finally, there were significant bidirectional cross-lagged associations between napping duration and MetS severity score after adjusting for all covariates. Root mean square residual (RMR) and comparative fit index (CFI) were 0.005 and 0.999, respectively, indicating a good fit to the observed data according to the criteria of RMR < 0.05 and CFI > 0.90. As shown in Figure 3 and Appendix A, the path coefficients shown are standardized. The path coefficient from baseline MetS score to follow-up napping duration (*β*, 0.037; *p* value, 0.004) was larger than that from baseline napping duration to follow-up MetS score (*β*, 0.025; *p* value, 0.026). However, the difference between these two coefficients was not statistically significant (Z value, 0.737; *p* value, 0.461).

## 4. Discussion

To the best of our knowledge, this is the first cohort study to explore the bidirectional associations between daytime napping and metabolic syndrome based on nationally representative data. Among middle-aged and older Chinese adults, compared with non-napping, longer daytime napping duration, especially >90 min/day, was significantly associated with a higher incidence of MetS at the four-year follow-up. In the subgroup analyses, our results demonstrated that excessive naps were significantly correlated with incident MetS only in individuals with enough night-time sleep or good sleep quality, and conversely, no significant associations were observed in those who have insufficient night-time sleep or relatively poor sleep quality. Among those diagnosed with MetS at baseline, extended naps were associated with lower remission rate of MetS only in adequate night-time sleepers. With respect to reverse associations, participants with MetS tended to nap longer during the follow-up period. In addition, hyperglycemia, hypertriglyceridemia, hypertension, and central obesity were all significantly associated with future excessive naps. Finally, a cross-lagged panel analysis further verified bidirectional relationship between longer naps and MetS.

Napping is traditionally believed to be a good habit. However, we found that longer naps may be detrimental to the cardiometabolic health. In the general population, our prospective evidence supports the conclusion of some published, cross-sectional studies that longer napping duration is associated with MetS [7,16,17,18,19]. Another cross-sectional study based on the same data source, CHARLS, also reported the similar research results [14]. A prospective cohort study of Spanish young individuals [15] demonstrated that average daytime napping > 30 min had a positive association with incidence of MetS, and our study extended this association to the middle-aged and older Chinese adults. Another longitudinal study found that longer napping duration was associated with a lower risk of MetS reversion in the general population [4]. However, such a significant association was found in our subgroup with ≥7 h of night-time sleep, rather than whole population, may be due to the between-study heterogeneity. Participants whose night-time sleep duration < 7 h accounted for 49.1% in our study and only 5.5% in their study [4]. This also indicated that napping may pay a more prominent adverse role in those with adequate night-time sleep. Our study revealed a J-curve relation between napping duration and the incidence of MetS, which is also consistent with previous studies [7,34]. An inverted J-curve relationship between nap and the remission of MetS in the adequate night-time sleepers was reported for the first time. The dose–response relationship further confirmed that a shorter nap was not significantly related to the development of MetS, and a significant association was only observed when the napping duration was longer than about 90 min/day.

Some potential mechanisms can explain the adverse effect of long daytime napping on MetS. We speculated that daytime napping can extend bedtime, so that the decreased thermogenesis and energy expenditure may result in abnormal metabolism of components of MetS mediated through obesity [25]. When people get enough and good night-time sleep, naps will further increase the bedtime to make the adverse effects more serious. In addition, circadian rhythm disturbances arising from excessive naps could be another explanation for this relationship, and circadian misalignment will sequentially result in metabolic and endocrine abnormalities and affect body hormones [7]. Several studies found that circadian disruption can lead to insulin resistance, obesity, hyperglycemia, and dyslipidemia [35,36,37]. Moreover, extended naps are related to elevated evening cortisol levels that might also increase the body’s resistance to insulin and lead to high levels of blood pressure [38]. Furthermore, excessive naps might be the clinical manifestations of underlying diseases. For example, individuals taking longer naps during the day were more likely to suffer from obstructive sleep apnea. The metabolic consequences of obstructive sleep apnea also may contribute to the incidence of MetS [39,40]. Finally, abnormal systemic inflammatory markers arising from extended naps and the elevation of blood pressure mediated by sympathetic activation induced by excessive naps could also be explanations [41,42].

Interestingly, the results of the subgroup analyses showed that longer daytime napping might have no adverse effect on MetS among individuals with severe sleep debt caused by short night-time sleep, poor sleep quality, or with depressive symptoms. Reduced sleep duration and quality can increase the risk of transition from metabolic health to metabolic unhealthy status [43,44,45]. A study reported that people can take naps to compensate for poor sleep quality or tiredness caused by short sleep, and that next-day nap duration increases as a result of poor sleep quality and short sleep duration [46]. Randomized controlled trials [11,12] found that daytime napping can reverse the effects of nocturnal sleep loss on cortisol, interleukin-6, and urinary norepinephrine secretion to normalize abnormal endocrine metabolism. This mechanism indicated that daytime napping could improve neuroendocrine stress, immune recovery, and cardiovascular health [12]. It is this compensation mechanism that can neutralize the harms of longer napping to weaken the adverse metabolic consequences caused by a longer napping duration. In short, the impact of daytime napping on metabolic health should consider the situation of nocturnal sleep. A previous study [13] reported that the association between naps and greater risk of mortality and cardiovascular events was only observed in those who slept at least 6 h at night but not in those with a shorter duration of nocturnal sleep, which is similar to our conclusion. Therefore, our results may provide a mechanism linking longer daytime napping duration and cardiovascular mortality via mediation of MetS.

Innovatively, our results also indicate a potential bidirectional relationship between MetS and daytime napping. MetS and its components also can change subsequent napping behavior. We can see that as the severity of MetS increases, the napping duration becomes increasingly longer during the follow-up period. Our cross-lagged analyses further strengthen this bidirectional relationship. In a cross-sectional study of adults among the French population, they found that subjects with overweight, obesity, hypertension, diabetes, depression, or anxiety disorders tended to have the habit of daytime napping compared with individuals without these disorders [47]. Similarly, when we analyzed each MetS components separately, hyperglycemia, hypertriglyceridemia, hypertension, and central obesity were all significantly associated with excessive naps. We speculated that people may develop the habit of excessive napping to cope with poor health and or fatigue caused by metabolic-related diseases. For instance, insulin resistance is central to the development of MetS. Altered insulin and glucose metabolism could lead to the loss of energy, which may cause fatigue and lead to longer napping duration [48]. Thus, excessive daytime sleepiness can be an early warning signal of MetS, enabling its early intervention to prevent or delay the onset of cardiometabolic disease.

### Strengths and Limitations

Our study had several major strengths. Innovatively, we evaluated the associations between daytime napping duration and the occurrence and remission of MetS in different subgroups stratified by night-time sleep duration and sleep quality. We also confirmed a bidirectional link between excessive daytime napping and MetS for the first time, based on this cohort study. A range of sensitivity analyses were conducted to strengthen our conclusions. The other strength of this study is the use of a large nationwide representative sample covering 28 provinces in mainland China, so these results can be generalized to the Chinese population.

However, several limitations need to be considered. Firstly, measurement of daytime napping duration, night-time sleep duration, sleep quality, and some health status indicators were based on self-reports and not objectively measured, which may lead to recall bias and misclassification of exposure. Additionally, although we adjusted for several potential confounders, other unmeasured or residual confounders may still affect our findings, such as obstructive sleep apnea syndrome. Of note, the adjusted variables in the models, sleep quality and night-time sleep duration, can reflect part of the problem of sleep disorders, but obstructive sleep apnea syndrome may not always be expressed as a subjective sleep disorder. Finally, our results in this single cohort study need further validation in other populations.

## 5. Conclusions

Among middle-aged and older Chinese adults, a bidirectional relationship was observed between daytime napping duration and MetS. On the one hand, longer daytime napping duration, especially >90 min/day, was significantly associated with both the higher incidence and lower remission rate of MetS except in those with less night-time sleep duration or poor sleep quality. On the other hand, patients with MetS at baseline tended to have subsequent longer napping duration. Our research provides the guidance and direction for appropriate daytime napping duration according to different night-time sleep duration and sleep quality to prevent MetS and improve cardiometabolic health. Moreover, monitoring napping duration to detect anomalies of napping behavior may aid the early detection and secondary prevention of MetS.

## Figures and Tables

**Figure 1 nutrients-14-05292-f001:**
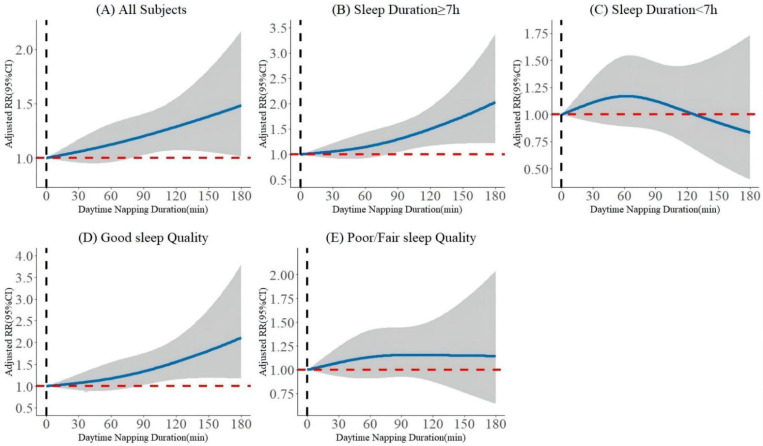
Longitudinal dose–response relationship between baseline daytime napping duration and the incidence of MetS. The curve was estimated by restricted cubic spline function with four knots. Solid lines indicate aRRs. The reference was set to 0 min. The shadow represents 95% confidence intervals.

**Figure 2 nutrients-14-05292-f002:**
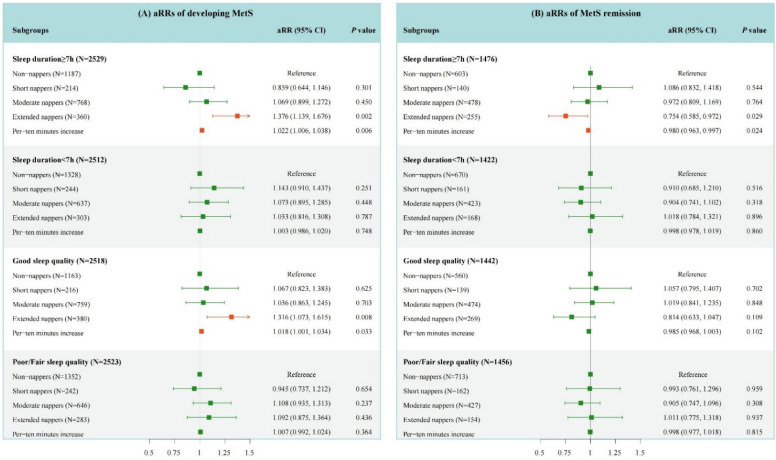
The association of daytime napping duration with the incidence and remission of MetS in the subgroups by night-time sleep duration and quality. Graphs show aRR and 95% CIs for the incidence of MetS (**A**), and the remission of MetS (**B**) adjusted for age, gender, living residence, marital status, educational level, smoking status, drinking status, depressive symptoms, night-time sleep quality, night-time sleep duration, SUA, LDL-C, and hsCRP. Napping duration of non-nappers is 0 min/day; Napping duration of short nappers is >0 min/day to ≤30 min/day; Napping duration of moderate nappers is >30 min/day to ≤90 min/day; Napping duration of extended nappers is >90 min/day. Orange means aRR value is statistically significant; green means aRR value is not statistically significant.

**Figure 3 nutrients-14-05292-f003:**
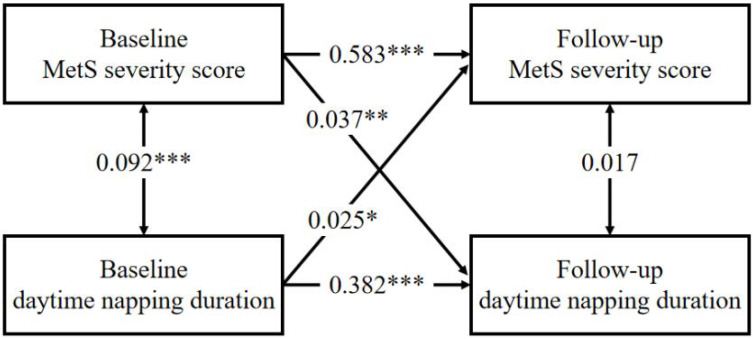
A cross-lagged panel analyses of MetS severity score and daytime napping duration at baseline and follow-up. Abbreviation: MetS severity score, the number of metabolic syndrome components. adjusted for: age, gender, living residence, marital status, educational level, smoking status, drinking status, depressive symptoms, night-time sleep quality, night-time sleep duration, serum uric acid, LDL-C, and hsCRP. Correlations are represented via double-headed arrows and regressive paths via single headed arrows. * *p* < 0.05; **, *p* < 0.01; ***, *p* < 0.001.

**Table 1 nutrients-14-05292-t001:** Baseline characteristics of the study population in three sub-cohorts.

Characteristics	Sub-Cohort 1	Sub-Cohort 2	Sub-Cohort 3
No. of participants	5041	2898	11,390
Age (years), mean (SD)	57.89 (9.12)	58.53 (8.77)	58.04 (9.30)
Male, *n* (%)	2711 (53.8)	1006 (34.7)	5426 (47.6)
Married, *n* (%)	4333 (86.0)	2458 (84.8)	9636 (84.6)
Elementary school or above, *n* (%)	2808 (55.7)	1496 (51.6)	6254 (54.9)
Rural residence, *n* (%)	4337 (86.0)	2345 (80.9)	9362 (82.2)
Smoking status, *n* (%)			
Current smoker	1832 (36.3)	656 (22.6)	3584 (31.5)
Former smoker	390 (7.7)	199 (6.9)	838 (7.4)
Non-smoker	2819 (55.9)	2043 (70.5)	6968 (61.2)
Drinking status, *n* (%)			
More than once a month	1490 (29.6)	578 (19.9)	3000 (26.3)
Drink but less than once a month	417 (8.3)	223 (7.7)	888 (7.8)
Never	3134 (62.2)	2097 (72.4)	7502 (65.9)
Physical activity, *n* (%)			
None	521 (10.3)	388 (13.4)	1407 (12.4)
Mild	1001 (19.9)	795 (27.4)	2433 (21.4)
Moderate	1520 (30.2)	936 (32.3)	3424 (30.1)
Vigorous	1999 (39.7)	779 (26.9)	4126 (36.2)
Depressive symptoms, *n* (%)	1863 (37.0)	1030 (35.5)	3705 (34.4)
BMI (kg/m^2^), mean (SD)	23.10 (4.34)	25.83 (3.76)	23.63 (4.31)
SUA (mg/dL), mean (SD)	4.29 (1.24)	4.58 (1.52)	4.42 (1.23)
HsCRP (mg/L), median [IQR]	0.87 [0.49, 2.07]	1.31 [0.68, 2.50]	0.99 [0.54, 2.12]
LDL-C (mg/dL), mean (SD)	99.51 (40.41)	131.84 (54.20)	118.38 (35.40)
Antihypertensive agents, *n* (%)	349 (6.9)	855 (29.5)	1670 (14.7)
Hypoglycemic agents, *n* (%)	45 (0.9)	205 (7.1)	335 (2.9)
Lipid-lowering agents, *n* (%)	21 (0.4)	318 (11.0)	426 (3.7)
sleeping pills/anti-depressive treatment, *n* (%)	30 (0.6)	13 (0.4)	66 (0.6)
Good sleep quality, *n* (%)	2518 (50.0)	1442 (49.8)	5770 (50.7)
Night-time sleep duration (h), mean (SD)	6.40 (1.89)	6.45 (1.85)	6.43 (1.86)
Napping duration in 2011 (min/day), mean (SD)	36.29 (44.46)	40.27 (44.56)	37.37 (44.07)
Napping duration in 2011, *n* (%)			
0 min/day	2515 (49.9)	1273 (43.9)	-
≤30 min/day	458 (9.1)	301 (10.4)	-
30–90 min/day	1405 (27.9)	901 (31.1)	-
>90 min/day	663 (13.2)	423 (14.6)	-
With MetS at baseline, *n* (%)	0 (0.0)	2898 (100.0)	3936 (34.6)
Outcome variables			
The incidence of MetS, *n* (%)	1126 (22.3)	-	-
The reversion of MetS, *n* (%)	-	828 (28.6)	-
Napping duration in 2013 (min/day), mean (SD)	-	-	42.34 (46.30)
Napping duration in 2015 (min/day), mean (SD)	-	-	42.39 (46.11)

Data are presented as the mean (SD), median [IQR] or number (%), as appropriate. Abbreviations: SD, standard deviation; IQR, interquartile range; BMI, body mass index; MetS, metabolic syndrome; SUA, serum uric acid; LDL-C, low-density lipoprotein cholesterol; hsCRP, high sensitivity C-reactive protein.

**Table 2 nutrients-14-05292-t002:** Longitudinal associations of baseline daytime napping duration with the occurrence and remission of MetS.

	N	Case, *n* (%)	Model 1		Model 2	
			aRR (95% CI)	*p*	aRR (95% CI)	*p*
Occurrence of MetS (N = 5041)						
per-ten minutes increase			1.014 (1.003, 1.026)	**0.012**	1.013 (1.002, 1.024)	**0.027**
non-nappers ^a^	2515	556 (22.1)	Reference	-	Reference	-
short nappers ^a^	458	105 (22.9)	1.033 (0.863, 1.237)	0.726	1.008 (0.843, 1.205)	0.933
moderate nappers ^a^	1405	300 (21.4)	1.087 (0.961, 1.230)	0.185	1.072 (0.946, 1.215)	0.275
extended nappers ^a^	663	165 (24.9)	1.242 (1.071, 1.441)	**0.004**	1.216 (1.047, 1.413)	**0.011**
Remission of MetS (N = 2898)						
per-ten minutes increase			0.989 (0.976, 1.002)	0.094	0.991 (0.978, 1.005)	0.203
non-nappers ^a^	1273	370 (29.1)	Reference	-	Reference	-
short nappers ^a^	301	90 (29.9)	1.013 (0.835, 1.230)	0.894	1.029 (0.847, 1.251)	0.773
moderate nappers ^a^	901	257 (28.5)	0.949 (0.829, 1.086)	0.445	0.971 (0.848, 1.111)	0.667
extended nappers ^a^	423	111 (26.2)	0.867 (0.722, 1.040)	0.123	0.892 (0.741, 1.073)	0.224

Abbreviations: CI, confidence interval; aRR, adjusted relative risk; MetS, metabolic syndrome; LDL-C, low-density lipoprotein cholesterol; SUA, serum uric acid; hsCRP, high sensitivity C-reactive protein. ^a^ Daytime napping duration of non-nappers: 0 min/day; Daytime napping duration of short nappers: >0 min/day to ≤30 min/day; Daytime napping duration of moderate nappers: >30 min/day to ≤90 min/day; Daytime napping duration of extended nappers: >90 min/day. Model 1: adjusted for age and gender. Model 2: adjusted for age, gender, living residence, marital status, educational level, smoking status, drinking status, depressive symptoms, night-time sleep quality, night-time sleep duration, SUA, LDL-C, and hsCRP. Bold *p* value denotes statistical significance (*p* < 0.05).

**Table 3 nutrients-14-05292-t003:** Longitudinal associations of baseline metabolic syndrome status and its components with follow-up daytime napping duration.

	N	Model 1		Model 2	
		*β* (95% CI)	*p* Value	*β* (95% CI)	*p* Value
Baseline MetS status					
Without MetS	7454	Reference		Reference	
With MetS	3936	5.081 (3.524, 6.637)	**<0.001**	2.745 (1.360, 4.130)	**<0.001**
Number of MetS components (MetS severity score)					
0 component	1102	Reference		Reference	
1 component	3204	−0.289 (−2.984, 2.406)	0.834	0.281 (−2.122, 2.683)	0.819
2 components	3148	2.411 (−0.306, 5.129)	0.082	2.047 (−0.380, 4.474)	0.098
3 components	2192	4.572 (1.704, 7.440)	**0.002**	2.837 (0.266, 5.408)	**0.031**
4 components	1160	6.454 (3.212, 9.696)	**<0.001**	4.032 (1.146, 6.918)	**0.006**
5 components	584	10.608 (6.533, 14.683)	**<0.001**	7.053 (3.435, 10.670)	**<0.001**
Baseline MetS components status					
Without hyperglycemia	7018	Reference		Reference	
With hyperglycemia	4372	3.897 (2.382, 5.412)	**<0.001**	2.236 (0.903, 3.570)	**0.001**
Without hypertriglyceridemia	8666	Reference		Reference	
With hypertriglyceridemia	2724	3.393 (1.682, 5.103)	**<0.001**	2.426 (0.903, 3.948)	**0.002**
Without low HDL-C	4759	Reference		Reference	
With low HDL-C	6631	0.989 (−0.515, 2.494)	0.197	0.278 (−1.062, 1.618)	0.684
Without hypertension	6126	Reference		Reference	
With hypertension	5264	2.742 (1.238, 4.246)	**<0.001**	1.967 (0.633, 3.301)	**0.004**
Without central obesity	6745	Reference		Reference	
With central obesity	4645	5.368 (3.795, 6.940)	**<0.001**	2.710 (1.309, 4.111)	**<0.001**

Abbreviations: CI, confidence interval; MetS, metabolic syndrome; *β*, regression coefficient; LDL-C, low-density lipoprotein cholesterol; hsCRP, high sensitivity C-reactive protein; SUA, serum uric acid; HDL-C, high-density lipoprotein cholesterol. Model 1: adjusted for age and gender. Model 2: adjusted for age, gender, living residence, marital status, educational level, smoking status, drinking status, depressive symptoms, night-time sleep quality, night-time sleep duration, SUA, LDL-C, hsCRP and napping duration at baseline in 2011. Bold *p* value denotes statistical significance (*p* < 0.05).

## Data Availability

The data that support the findings of this study are available in China Health and Retirement Longitudinal Study (CHARLS), at http://charls.pku.edu.cn/ (accessed on 19 July 2021). Materials are available on request to the corresponding author.

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
