# Peer review of "Bidirectional Associations between Daytime Napping Duration and Metabolic Syndrome: A Nationally Representative Cohort Study"

_nutrients, 2022, doi:10.3390/nu14245292_

Round 1

Reviewer 1 Report

Review for the manuscript entitled Bidirectional associations between daytime napping duration and metabolic syndrome: a nationally representative cohort study.

The current study reveals a causal relationship in which daytime napping duration contributes to the development of MetS, while MetS also prolongs daytime napping duration. While the analysis is meticulous, the story is clear and easy to understand. 

The reviewer would like to make only the following suggestion:

Is it possible that sleep apnea syndrome (SAS) may partially intervene in the relationship between daytime napping duration and MetS?

That is, people with SAS also require long naps, and SAS itself may be a potential risk for MetS (J Clin Sleep Med. 2007 Aug 15;3(5):467-72.) In addition, SAS may not always be expressed as a subjective sleep disorder.

The reviewer therefore recommends adding findings on SAS in Discussion.

Author Response

Dear reviewer,

Thanks for the valuable comments concerning our manuscript entitled “Bidirectional associations between daytime napping duration and metabolic syndrome: a nationally representative cohort study” (ID: 2025494). We have responded to the comments, and revised the manuscript accordingly. Please see the attachment.

Sincerely yours

Lixin Tao, PhD, Associated Professor

School of Public Health

Capital Medical University

Beijing100069, PR China

Tel/fax: +86-10-8391-1778

Reviewer 2 Report

Bidirectional associations between daytime napping duration and metabolic syndrome: a nationally representative cohort study 

This is a very interesting work in which authors try to to examine the bidirectional associations between daytime napping duration and metabolic syndrome (MetS) using data from China Health and Retirement Longitudinal Study. The paper is well structured and displays insight into the situation. It addresses an important issue not only substantial and relevant for clinical placements and physicians but also for the fields of primary and secondary prevention.  

Overall it is an excellent work, I have some minor comments:

1) Even though the methodology section is adequate, I would suggest to separate it with more subheading in order to be easier for the reader. For example the bioethical approvement and the covariate section.

Author Response

Dear reviewer,

Thanks for the valuable comments concerning our manuscript entitled “Bidirectional associations between daytime napping duration and metabolic syndrome: a nationally representative cohort study” (ID: 2025494). We are also very grateful for your recognition of our work. We have responded to the comments and revised the manuscript accordingly. Please see the attachment.

Sincerely yours

Lixin Tao, PhD, Associated Professor

School of Public Health

Capital Medical University

Beijing100069, PR China

Tel/fax: +86-10-8391-1778
